# Facemasks: Perceptions and use in an ED population during COVID-19

Vidya Eswaran[1,2]*, Anna Marie Chang[3], R. Gentry Wilkerson[4], Kelli N. O'Laughlin[5], Brian Chinnock[1], Stephanie A. Eucker[6], Brigitte M. Baumann[7], Nancy Anaya[1], Daniel G. Miller[8], Adrianne N. Haggins[9], Jesus R. Torres[10,11], Erik S. Anderson[12], Stephen C. Lim[13], Martina T. Caldwell[14], Ali S. Raja[15], Robert M. Rodriguez[1], The REVVED-UP Investigators[¶]

1 Department of Emergency Medicine, University of California, San Francisco, CA, United States of America, 2 National Clinician Scholars Program, Philip R Lee Institute of Health Policy Studies, University of California, San Francisco, CA, United States of America, 3 Department of Emergency Medicine, Thomas Jefferson University, Sidney Kimmel Medical College, Philadelphia, PA, United States of America, 4 Department of Emergency Medicine, University of Maryland School of Medicine, Baltimore, MD, United States of America, 5 Department of Emergency Medicine and Global Health, University of Washington, Seattle, WA, United States of America, 6 Division of Emergency Medicine, Department of Surgery, Duke University School of Medicine, Durham, NC, United States of America, 7 Department of Emergency Medicine, Cooper Medical School of Rowan University, Cooper University Hospital, Camden, NJ, United States of America, 8 Departments of Emergency and Internal Medicine, University of Iowa Hospitals and Clinics, Iowa, IA, United States of America, 9 Department of Emergency Medicine, University of Michigan, Ann Arbor, Michigan, 10 Department of Emergency Medicine, Olive View UCLA Medical Center, University of California Los Angeles Schools of Medicine, Los Angeles, CA, United States of America, 11 National Clinician Scholars Program, University of California, Los Angeles, CA, United States of America, 12 Department of Emergency Medicine, Alameda Health System, Oakland, CA, United States of America, 13 Section of Emergency Medicine, University Medical Center New Orleans, Louisiana State University Health Sciences Center, New Orleans, LA, United States of America, 14 Department of Emergency Medicine, Henry Ford Hospital, Detroit, MI, United States of America, 15 Department of Emergency Medicine, Massachusetts General Hospital, Boston, MA, United States of America

¶ The complete membership of the REVVED-UP Investigators can be found in the Acknowledgments.
* vidya.eswaran@ucsf.edu

**Data Availability Statement:** Data files are available from the Dryad repository at: https://datadryad.org/stash/share/0j3mkknTmAQvvMrhbGT3LU4uizhjFaPltnOlebtxPRU.

## Abstract

### Study objective

Facemask use is associated with reduced transmission of SARS-CoV-2. Most surveys assessing perceptions and practices of mask use miss the most vulnerable racial, ethnic, and socio-economic populations. These same populations have suffered disproportionate impacts from the pandemic. The purpose of this study was to assess beliefs, access, and practices of mask wearing across 15 urban emergency department (ED) populations.

### Methods

This was a secondary analysis of a cross-sectional study of ED patients from December 2020 to March 2021 at 15 geographically diverse, safety net EDs across the US. The primary outcome was frequency of mask use outside the home and around others. Other outcome measures included having enough masks and difficulty obtaining them.

### Results

Of 2,575 patients approached, 2,301 (89%) agreed to participate; nine had missing data pertaining to the primary outcome, leaving 2,292 included in the final analysis. A total of

**Funding:** The author(s) received no specific funding for this work.

**Competing interests:** The authors have declared that no competing interests exist.

79% of respondents reported wearing masks "all of the time" and 96% reported wearing masks over half the time. Subjects with PCPs were more likely to report wearing masks over half the time compared to those without PCPs (97% vs 92%). Individuals experiencing homelessness were less likely to wear a mask over half the time compared to those who were housed (81% vs 96%).

## Conclusions

Study participants reported high rates of facemask use. Respondents who did not have PCPs and those who were homeless were less likely to report wearing a mask over half the time and more likely to report barriers in obtaining masks. The ED may serve a critical role in education regarding, and provision of, masks for vulnerable populations.

## Introduction

The role of face masks in the control of the 2019 Sars-COV-2 (COVID-19) pandemic has generated considerable controversy [1]. In the United States, one of the first recommendations regarding masking came from the Surgeon General in February of 2020, who urged Americans to avoid purchasing masks, out of fear of shortages for healthcare providers [2]. This was followed in April 2020 by recommendations from the Centers for Disease Control and Prevention to wear cloth face covers in public areas [3]. Since then, there has been a plethora of evidence describing the benefit of mask use in curbing the spread of COVID-19 [4].

The COVID-19 pandemic has had a disproportionate impact on historically marginalized groups including racial and ethnic minority populations [5, 6] and the socioeconomically disadvantaged [7]. For example, individuals experiencing homelessness were found to have 1.3x higher case fatality rate than the general population [8]. The emergency department (ED) is considered society's "safety net" [9], and is often the only point of health care access for many marginalized populations [10]. Other surveys have shown that negative attitudes toward masks, and mask mandates, persist [11]. As demonstrated in previous work on vaccine hesitancy [12], online and phone recruitment survey methods may fail to sample a number of vulnerable populations who are at high risk of complications from COVID-19, such as the unhoused, those experiencing financial hardship and minoritized populations [5].

The objective of this study was to assess mask wearing beliefs and practices, specifically mask use, access to masks, and reasons for not wearing masks, among patients treated in emergency departments that serve as critical points of health care access and intervention for several vulnerable populations.

## Materials and methods

### Study design and setting

We conducted this secondary analysis of a previously published study regarding ED patients perceptions' of COVID-19 vaccination [12]. The parent study was a prospective, cross-sectional survey of ED patients at 15 safety net EDs in 14 US cities. The University of California Institutional Review Board approved this study. Verbal consent was obtained. We follow the Strengthening the Reporting of Observational Studies in Epidemiology (STROBE) [13] guidelines in this manuscript.

## Survey design

We adapted survey questions from published instruments [14]. We reviewed the survey instrument with the UCSF COVID-19 Patient Community Advisory Board. We subsequently pilot tested the instrument on six patients at the primary site and found excellent comprehension and response consistency [12]. (S1 File).

## Selection of participants

Sites enrolled patients between December 20, 2020, and March 7, 2021, with a goal of enrolling 150 adult ED patients over the study period. Exclusion criteria were major trauma, transfer from other facilities, incarceration, psychiatric hold, intoxication, altered mental status, critical illness, and temporary visit from other countries. At 11 sites, surveys were conducted in-person. Due to pandemic-related constraints, three sites used phones to call into patients' ED rooms, and one site called patients immediately after discharge.

Participant ethnicity (Latinx/non-Latinx) and race were self-reported. We categorized those who self-identified as any race other than Latinx as 'reported race', non-Latinx (i.e. Black, non-Latinx and White, non-Latinx). If the patient identified themselves as Latinx, they were placed in that category and not in that of any other race. If an individual identified as more than one non-Latinx race, they were categorized as multiracial.

Individuals who reported that they were currently applying for health insurance, were unsure if they were insured, or if their response to the question was missing (18 respondents) were categorized as uninsured in a binary variable, and separate analysis was done based on type of insurance reported. The survey submitted in our (S1 File) is the version used at the lead site. Each of the remaining sites revised their survey to include wording applicable to their community (i.e., the site in Los Angeles changed Healthy San Francisco to Healthy Los Angeles), and these local community health plans were coded together.

We identified individuals who reported English and Spanish as their primary language, and grouped those who reported Arabic, Bengali, Cantonese, Tagalog, or Other as "Other" primary language. With regards to gender, we categorized those who identified as gender queer, nonbinary, trans man and trans woman as "other".

## Study outcomes and key variables

Our primary outcome was subjects' response to the question, "Do you wear a mask when you are outside of your home when you are around other people?" with answer choices a) always, b) most of the time (more than 50%), c) sometimes, but less than half of the time (less than 50%), and d) I never wear a mask. Respondents were provided with these percentages to help quantify their responses. We stratified respondents into two groups: those who responded always or most of the time as "wears masks over half the time" and those who responded sometimes or never as "wears masks less than half the time. Other outcomes included reasons, when applicable, individuals did not wear masks as well responses to queries regarding from where patients obtained their masks, difficulty obtaining masks, for which we assessed for differences by race, having a PCP, and usual source of healthcare.

We sorted each of the 15 sites into four geographic regions within the United States. There were 3 sites located in New Jersey, Massachusetts, and Pennsylvania which we categorized in the Northeast region. We categorized 3 sites in Michigan and Iowa as Midwest, and 3 sites in North Carolina, Louisiana, and Maryland as the South. There were 6 sites located on the West Coast from California and Washington State.

## Data analysis

We summarized demographics as counts and frequency percentages with 95% confidence intervals (CIs), including nonresponses to individual questions. We performed separate tests of proportions to assess differences in wearing masks over half the time for each of group (gender, race, hospital region, hospital, primary language, homelessness, insurance type, having a PCP, usual source of healthcare, and prior COVID-19 diagnosis) as independent variables for the outcome. We calculated risk differences with 95% CIs for responses to questions regarding source of masks and difficulty in obtaining masks by having a PCP, usual source of care and race. We managed data using REDCap and conducted analyses using Stata v. 16.1 (StataCorp LLC, College Station, TX).

## Results

### Characteristics of study subjects

Of the 2,575 patients approached, 2,301 (89%) agreed to participate; nine had missing answers for the key outcome (frequency of mask wearing outside the home) leaving 2,292 individuals included in this study. Their median age was 48 years (interquartile range [IQR] 34–61), and the race/ethnicity composition was 23% Latinx, 37% White, non-Latinx, 29% Black, non-Latinx, and 4% Asian. Only 15% reported having been previously diagnosed with COVID-19. The majority (81%) reported having a primary care physician (PCP) or clinic. The ED was the usual source of health care for 12% of respondents, and 64% of those were without a PCP.

### Mask use

Most (79%) reported wearing masks "all of the time," and 96% reported wearing masks over half the time (S1 Table). Subjects with primary care providers were more likely to report wearing masks over half the time as compared to subjects without PCPs (97% vs 92%, difference 4%, 95% CI 1%-7%) (Table 1). White, non-Latinx, respondents were slightly less likely than Black, non-Latinx, Latinx, and Asian, non-Latinx patients to report wearing a mask over half the time (95% vs 96%, 97% and 99%; p<0.01). Individuals experiencing homelessness were

**Table 1. Demographic information by mask wearing frequency and 95% CI.**

| Do you wear a mask when you are outside your home or around other people? | | | |
|---|---|---|---|
| | **Total** | **Never or Sometimes** | **Mostly or Always** |
| n (%) | 2,292 | 95 (4%) | 2,197 (96%) |
| Age (mean(SD)) | 48 (17) | 46 (18) | 48 (17) |
| Gender, (n(%, 95% CI)) | | | |
| Female | 1,129 | 28 (2%, 2%-4%) | 1,101 (98%, 96%-98%) |
| Male | 1,146 | 65 (6%, 4%-7%) | 1,081 (94%, 93%-95%) |
| Other | 8 | 1 (12.5%, 2%-54%) | 7 (87.5%, 46%-98%) |
| Missing | 9 | 1 (11%, 1%-50%) | 8 (89%, 50%-98%) |
| Race, (n(%, 95% CI)) | | | |
| Latinx | 535 | 15 (3%, 2%-5%) | 520 (97%, 95%-98%) |
| White, non-Latinx | 853 | 46 (5%, 4%-7%) | 807 (95%, 93%-96%) |
| Black, non-Latinx | 667 | 29 (4%, 3%-6%) | 638 (96%, 94%-97%) |
| Asian, non-Latinx | 94 | 1 (1%, 0.1%-7%) | 93 (99%, 93%-100%) |
| Middle Eastern, non-Latinx | 26 | 0 (0%) | 26 (100%) |
| Native American, non-Latinx | 12 | 3 (25%, 8%-55%) | 9 (75%, 45%-92%) |
| Native Hawaiian, non-Latinx | 6 | 0 (0%) | 6 (100%) |
| Other | 34 | 0 (0%) | 34 (100%) |

*(Continued)*

**Table 1.** (Continued)

**Do you wear a mask when you are outside your home or around other people?**

| | Total | Never or Sometimes | Mostly or Always |
|---|---|---|---|
| Multiracial | 41 | 0 (0%) | 41 (100%) |
| Decline | 12 | 0 (0%) | 12 (100%) |
| Missing | 12 | 1 (8%, 1%-41%) | 11 (92%, 59%-99%) |
| Homeless, (n(%, 95% CI)) | | | |
| Yes | 84 | 16 (19%, 12%-29%) | 68 (81%, 71%-99%) |
| Insured, (n(%, 95% CI)) | | | |
| Yes | 1,998 | 82 (4%, 3%-5%) | 1916 (96%, 95%-97%) |
| No | 294 | 13 (4%, 3%-7%) | 281 (96%, 92%-97%) |
| Insurance Type, (n(%, 95% CI)) | | | |
| Private | 871 | 17 (2%, 2%-3%) | 854 (98%, 97%-99%_ |
| VA | 13 | 1 (8%, 1%-39%) | 12 (92%, 61%-99%) |
| ACA | 99 | 13 (13%, 8%-21%) | 87 (97%, 79%-92%) |
| Medicare | 377 | 18 (5%. 3%-7%) | 359 (95%, 92%-97%) |
| Medicaid | 540 | 28 (5%. 4%-7%) | 512 (95%, 93%-96%) |
| Kaiser | 20 | 1 (5%, 0.7%-28%) | 19 (95%, 72%-99%) |
| Local Community Health Plan | 31 | 0 (0%) | 31 (100%) |
| Uninsured | 278 | 14 (5%, 3%-8%) | 264 (95%, 92%-97%) |
| Unsure | 45 | 1 (2%, 0.3%-14%) | 44 (95%, 92%-97%) |
| Missing | 18 | 2 (11%, 3%-35%) | 16 (98%, 86%-100%) |
| Hospital Region, (n(%, 95% CI)) | | | |
| North East | 457 | 14 (3%, 2%-5%) | 443 (97%, 95%-98%) |
| Midwest | 451 | 25 (5%, 4%-8%) | 426 (94%, 92%-96%) |
| South | 444 | 25 (6%, 4%-8%) | 419 (94%, 92%-96%) |
| West Coast | 939 | 31 (3%, 2%-5%) | 908 (97%, 95%-98%) |
| Missing | 1 | 0 (0%) | 1 (100%) |
| Primary Language, (n(%, 95% CI)) | | | |
| English | 1,848 | 87 (5%, 4%-6%) | 1761 (95%, 94%-96%) |
| Spanish | 344 | 7 (2%, 0.1%-4%) | 337 (98%, 96%-99%) |
| Other | 95 | 0 (0%) | 95 (100%) |
| Missing | 5 | 1 (20%, 3%-69%) | 4 (80%, 31%-97%) |
| Have a Primary Care Physician, (n(%, 95% CI)) | | | |
| No | 423 | 32 (7%, 5%-10%) | 391 (92%, 89%-95%) |
| Yes | 1,859 | 53 (3%, 3%-4%) | 1,796 (97%, 96%-97%) |
| Missing | 10 | 0 (0%) | 10 (100%) |
| Usual Source of Care, (n(%, 95% CI)) | | | |
| Primary Care Physician | 1,859 | 63 (3%, 3%-4%) | 1796 (97%, 96%-97%) |
| ED | 274 | 26 (9%, 6%-13%) | 248 (90%, 86%-93%) |
| Clinic | 60 | 1 (2%, 0.2%-11%) | 59 (98%, 89%-100%) |
| Urgent Care | 51 | 1 (2%, 0.3%-13%) | 50 (98%, 87%-100%) |
| Unsure | 25 | 1 (4%, 0.5%-23%) | 24 (96%, 76%, 99%) |
| Other | 23 | 3 (13%, 4%-33%) | 20 (87%, 66%-96%) |
| Previous COVID-19 Diagnosis, (n(%, 95% CI)) | | | |
| No | 1,910 | 83 (4%, 3%-5%) | 1827 (96%, 95%-96%) |
| Yes | 341 | 10 (3%, 1%-5%) | 331 (97%, 95%-98%) |
| Unsure | 38 | 2 (5%, 1%-19%) | 36 (95%, 81%-99%) |
| Missing | 3 | 0 (0%) | 3 (100%) |

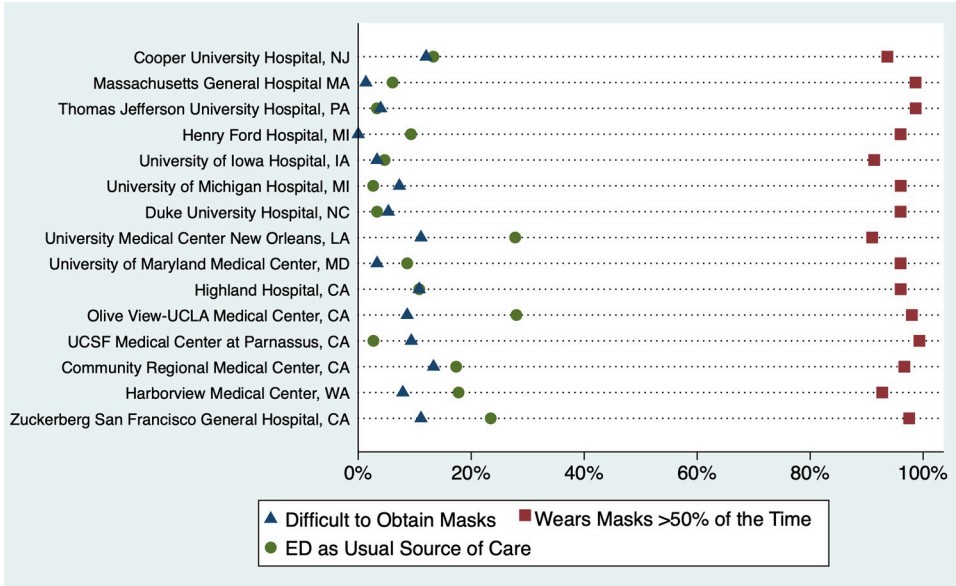

**Fig 1. Percentage of respondents by Hospital reporting difficulty obtaining masks, the ED as usual source of care and whether they wear masks over half the time.**

less likely to wear a mask over half the time compared to those who were housed (81% vs 96%, difference 15%, 95% CI 7–24%). Males were also less likely to wear a mask over half the time compared to females (94% vs 97%, difference 3%, 95% CI 2–5%). There was no significant difference in mask wearing between hospital regions with 97% of respondents at Northeast hospitals, 94% at Midwest, 94% in the South and 97% on the West Coast reporting wearing masks over half the time (p = 0.101) (Fig 1).

Of the fourteen individuals (0.6%) who reported never wearing a mask, four reported that masks made it hard to breathe, four believed that masks do not work, three stated they were uncomfortable, and one each cited: tired of wearing masks, forget to wear them, just don't want to wear masks, and live in a facility where masks are not required.

## Obtaining masks

Respondents reported obtaining masks from grocery stores or pharmacies (57%), ordering them online (13%), and from friends or family (11%). Only 5% obtained masks from their doctor or clinic and 3% from EDs. Respondents with PCPs reported less difficulties obtaining masks compared to those without PCPs (6% vs 13%: difference 7%, 95% CI 3–10%). Those who used EDs as their primary source of care were more likely to report difficulty in obtaining masks compared to those who considered their PCP their primary source of care (15% vs 6%: difference 8%, 95% CI 4–13%). Subjects with a PCP were less likely to get masks from a shelter or food bank as compared to those without a PCP (1% vs 4%: difference 3%, 95% CI 1–5%). White, non-Latinx patients were less likely to obtain masks from grocery stores or pharmacies than Black, non-Latinx, or Latinx patients (48% vs 61% vs 69%, p<0.01), and were more likely to obtain them online (18% vs 10% vs 8%, p<0.01).

## Discussion

In this real-world ED study of diverse, urban populations, respondents reported a high rate of facemask use. These rates are higher than some which have been previously

reported [15]. However, the sources of masks and reasons for not wearing masks were similar to those found in other research [11]. Small but significant differences in mask wearing habits were found between those of different racial/ethnic groups, and between males and females, differences which have been found in prior surveys as well [16]. Our study adds to these data due to the unique population studied, patients in safety-net EDs. As such we were able to note that individuals experiencing homelessness were significantly less likely to wear masks most of the time than those who are housed. This potentially points to an unmet need amongst this population to provide education and supplies. With businesses such as coffeehouses closed, it is possible that access to electrical outlets was limited, making it difficult to charge mobile phones and other devices, and thus essential information regarding social distancing and protective measures may not have been easily transmitted to this group [17]. Mask wearing may have also been limited by inabilities to keep masks clean and dry or limited access to masks at food banks and shelters, especially early in the pandemic [18].

The primary limitation of this study is that we conducted this survey in December 2020 through February 2021 –notably not during the Delta or Omicron variant surges. With progressive "pandemic fatigue" [19] and decreased mask mandates [20], mask wearing practice may have changed. Because this data is self-reported, social desirability bias may have influenced respondents to answer in ways they thought may please survey administrators, even if not representative of their true behavior. Spectrum bias limits the generalizability of our findings: all sites were in urban areas affiliated with academic medical institutions, and we could only survey people who presented to EDs, thus excluding patients in rural areas, where mask use is known to be less prevalent [21, 22], and those who may have greater mistrust of health care systems. We did not separately assess race and ethnicity (S1 File and S1 Table) and thus it is possible that we did not accurately capture these demographics, especially for Latinx individuals. Nuances of mask use, such as correct positioning, were not assessed, and thus our results may overestimate effective mask wearing. Three sites used different methods for the survey (two via telephone during ED stay and one via telephone after ED discharge), formats which may lead to different answers to key questions.

Despite these limitations, this study delineates an opportunity for public health intervention from the ED–one that should not take significant time or cost. Our study shows that currently very few respondents received masks from the ED or their doctors' offices. Emergency departments should consider social determinants of health as they relate to the provision of masks to vulnerable populations, especially people experiencing homelessness [23, 24], ED providers should ask patients whether they wear masks and if they have enough of them, as well provide education regarding their use. Educating patients about the benefits of masks and providing masks to those who do not have enough should become part of provider care in the ED–perhaps as a standard component of triage or discharge. Further research, with more contemporary data, to assess the feasibility and acceptability among patients and providers of such interventions should be done to guide next steps.

## Conclusions

Our findings inform interventions in the ED focusing on homeless persons and those without PCPs, who had significantly lower rates of mask wearing and greater difficulty obtaining masks. Having a PCP may provide opportunities for education and distribution of facemasks, though it is also possible that accessing primary care is correlated with greater health care access and economic security, and thus greater access to masks.

## Supporting information

**S1 File. Survey instrument.**
(DOCX)

**S1 Table. Demographic information by mask wearing frequency.**
(DOCX)

## Acknowledgments

The authors would like to thank the REVVED-UP Investigators for their support throughout study implementation and specifically for their efforts with respect to data collection and manuscript review. The REVVED-UP Investigators include the authors of this study as well as Karen Adams, Kurt Auville, Nicole Byl, Bhanu Chadalawada, Benjamin Coleman, Alex Farthing, Joseph Graterol, Alaina Hunt, Morgan Kelly, Kyra Lasko, Sophie C. Morse, Graham Nichol, Berenice Perez, Evan Rusoja, Lindsey Shughart, Breena R. Taira, and Joshua R Tiao. The lead author of the REVVED-UP Investigators is Dr. Robert Rodriguez who may be contacted at robert.rodriguez@ucsf.edu.

## Author Contributions

**Conceptualization:** Anna Marie Chang, R. Gentry Wilkerson, Kelli N. O'Laughlin, Brian Chinnock, Stephanie A. Eucker, Brigitte M. Baumann, Nancy Anaya, Daniel G. Miller, Adrianne N. Haggins, Jesus R. Torres, Erik S. Anderson, Stephen C. Lim, Martina T. Caldwell, Ali S. Raja, Robert M. Rodriguez.

**Data curation:** Vidya Eswaran.

**Formal analysis:** Vidya Eswaran, Robert M. Rodriguez.

**Investigation:** Anna Marie Chang, R. Gentry Wilkerson, Kelli N. O'Laughlin, Brian Chinnock, Stephanie A. Eucker, Brigitte M. Baumann, Nancy Anaya, Daniel G. Miller, Adrianne N. Haggins, Jesus R. Torres, Erik S. Anderson, Stephen C. Lim, Martina T. Caldwell, Ali S. Raja, Robert M. Rodriguez.

**Methodology:** Vidya Eswaran, Anna Marie Chang, R. Gentry Wilkerson, Kelli N. O'Laughlin, Brian Chinnock, Stephanie A. Eucker, Brigitte M. Baumann, Nancy Anaya, Daniel G. Miller, Adrianne N. Haggins, Jesus R. Torres, Erik S. Anderson, Stephen C. Lim, Martina T. Caldwell, Ali S. Raja, Robert M. Rodriguez.

**Project administration:** Robert M. Rodriguez.

**Supervision:** Robert M. Rodriguez.

**Visualization:** Vidya Eswaran.

**Writing – original draft:** Vidya Eswaran, Robert M. Rodriguez.

**Writing – review & editing:** Vidya Eswaran, Anna Marie Chang, R. Gentry Wilkerson, Kelli N. O'Laughlin, Brian Chinnock, Stephanie A. Eucker, Brigitte M. Baumann, Nancy Anaya, Daniel G. Miller, Adrianne N. Haggins, Jesus R. Torres, Erik S. Anderson, Stephen C. Lim, Martina T. Caldwell, Ali S. Raja, Robert M. Rodriguez.

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
