## [Decision Letter · Decision Letter 0]

25 Feb 2022

PONE-D-22-02590Facemasks: Perceptions and Use in an ED Population During COVID-19PLOS ONE

Dear Dr. Eswaran,

Thank you for submitting your manuscript to PLOS ONE. After careful consideration, we feel that it has merit but does not fully meet PLOS ONE’s publication criteria as it currently stands. Therefore, we invite you to submit a revised version of the manuscript that addresses the points raised during the review process.

We look forward to receiving your revised manuscript.

Kind regards,

Francesca Baratta, PharmD, PhD

Academic Editor

PLOS ONE

Journal Requirements:

2. One of the noted authors is a group or consortium “The REVVED-UP”. In addition to naming the author group, please list the individual authors and affiliations within this group in the acknowledgments section of your manuscript. Please also indicate clearly a lead author for this group along with a contact email address.

Reviewers' comments:

Reviewer's Responses to Questions

**Comments to the Author**

1. Is the manuscript technically sound, and do the data support the conclusions?

Reviewer #1: Partly

Reviewer #2: Partly

2. Has the statistical analysis been performed appropriately and rigorously? 

Reviewer #1: I Don't Know

Reviewer #2: Yes

3. Have the authors made all data underlying the findings in their manuscript fully available?

Reviewer #1: Yes

Reviewer #2: Yes

4. Is the manuscript presented in an intelligible fashion and written in standard English?

Reviewer #1: Yes

Reviewer #2: Yes

5. Review Comments to the Author

Reviewer #1: The study presents an important aspect of mask use in vulnerable populations. There are a few considerations that could make the paper more robust and there are some questions needing clarification.

The introduction is quite short. Since the focus of the study was vulnerable populations, it would be helpful to include background on rates of COVID infections in this group of individuals compared to the general population or how the pandemic has impacted vulnerable groups differently and puts them at higher risk for COVID related complications.

The objective sentence should be clarified. The way it reads it seems as though it is specifically mask wearing in the ED, and not practices by individuals who were treated in the ED.

Since not all sites conducted the survey in the same manner (some while in the ED and some after discharge), this may be an additional limitation to note. Subjects may respond differently in different situations.

Not sure about the grouping of responses for mask wearing outside the home. This may be ok but would be helpful to include an additional table that shows frequency and percentage for each response of always, most of the time, sometimes and never. Always (100%) is quite different from most of the time (50-99%?). Were respondents provided the percentages (> or < 50% of the time) to help quantify their responses?

The reported mask wearing in the results is confusing. Again, would be helpful to provide numbers for each response and then group after that.

Survey item number 12 regarding regular clinic or doctor does not specifically state primary care doctor. Is it assumed that respondents understand this item meant primary care doctor? How would it differentiate if they saw a specialist regularly?

The health insurance survey response options are confusing. Did all 15 participating sites outside of the lead site offer MediCal, Kaiser or Healthy San Francisco? Each site probably has a similar version of these programs. Is Kaiser considered private insurance? How this data is reported should be clarified and may require to be grouped differently. Could also consider an overall grouping of insured and not insured and then provide the breakdown of different types.

On Table 1, results are presented in different ways but it is unclear which is what. Some are median and IQR, some appear to be frequency and percentage and others a range? Ok to report differently but each needs to be stated what is the measure. Otherwise it is confusing.

For the supplemental survey items, would consider only including the items that were relevant to the reported results for this study. For example, this study did not report on influenza or COVID vaccinations yet there are survey items included on this.

The Discussion could be more robust. More specifics on how this study compares to similar studies should be included. Should elaborate more on the issues presented, especially as related to vulnerable populations. Discussion should note the implications for clinical practice and areas for future or additional research.

The timeframe in which the study occured is stated but should be specifically noted it did not include the Delta or Omicron variant surges.

Additional study limitation is subject self reported data. There is always the possibility that respondents may not have answered truthfully but rather based on what they felt the healthcare team wanted to hear.

The Conclusion section should be more concise. This is typically a short summary of the overall findings. The second part of the conclusions regarding actionalble items and education belong in the discussion. Conclusion should not note anything new that wasn't already discussed.

Regarding the figure of percentage of respondents by hospital, it would also be helpful to include something similar with grouping by geographical region.

Overall the study is good and I commend the authors for their work. Improving on the points noted will help it to be more meaningful and contribute to the literature on this topic.

Reviewer #2: Thank you very much for allowing me to review the submission titled: Facemasks: Perceptions and Use in an ED Population During COVID-19, submitted by Karen Adams et all. This paper is a secondary analysis of an ED based study across the US focusing on safety net EDs in urban academic institutions. Their data is interesting but biased by where the study was run.

While they acknowledge that bias in their conclusion, throughout the result section we are able to see the degree of imbalance in their population studied. Their data shows that English was the primary language for 80%, with 81% having a PCP and only 12% being uninsured. By documenting that the white, non Latinx population buys their masks on line, we are left with the suggestion that these are adults of means, with access to on line ordering, a home/stable place of living for product to be delivered to and a credit card to pay the bill. In essence that does not translate to their results being extendable to match what the title of their study is. In general, their population is able to acquire masks and uses them as needed. Only a very few do not want or use masks, and even fewer document that they don’t want to use them. This does not match the large general population that are anti-mask, that uses social media to downplay the risks of COVID and play up the myth that masks are bad for you.

Perhaps we would have been better able to judge mask use if we looked at the general population that attended large (political) gatherings or casinos or even looked in states that never had a mask mandate.

6. PLOS authors have the option to publish the peer review history of their article (what does this mean?). If published, this will include your full peer review and any attached files.

Reviewer #1: No

Reviewer #2: No

---

## [Author Response · Author response to Decision Letter 0]

8 Mar 2022

We thank the Editor and reviewers for their insightful comments, which have greatly helped to improve our manuscript. We have addressed the comments in detail below.

Editor’s Comments:

Editor's Comments to the Author:

The introduction is quite short. Since the focus of the study was vulnerable populations, it would be helpful to include background on rates of COVID infections in this group of individuals compared to the general population or how the pandemic has impacted vulnerable groups differently and puts them at higher risk for COVID related complications.

Thank you for this comment, we have amended the introduction as you suggest, which can now be found on Page 3, Line 68, and is pasted here:

The role of face masks in the control of the 2019 Sars-COV-2 (COVID-19) pandemic has generated considerable controversy.[1] In the United States, one of the first recommendations regarding masking came from the Surgeon General in February of 2020, who urged Americans to avoid purchasing masks, out of fear of shortages for healthcare providers.[2] This was followed in April 2020 by recommendations from the Centers for Disease Control and Prevention to wear cloth face covers in public areas.[3] Since then, there has been a plethora of evidence describing the benefit of mask use in curbing the spread of COVID-19.[4] 

The COVID-19 pandemic has had a disproportionate impact on historically marginalized groups including racial and ethnic minority populations[5,6] and the socioeconomically disadvantaged.[7] For example, individuals experiencing homelessness were found to have 1.3x higher case fatality rate than the general population.[8] The emergency department (ED) is considered society’s “safety net”,[9] and is often the only point of health care access for many marginalized populations.[10] Other surveys have shown that negative attitudes toward masks, and mask mandates, persist.[11] As demonstrated in previous work on vaccine hesitancy,[12] online and phone recruitment survey methods may fail to sample a number of vulnerable populations who are at high risk of complications from COVID-19, such as the unhoused, those experiencing financial hardship and minoritized populations.[5] 

The objective sentence should be clarified. The way it reads it seems as though it is specifically mask wearing in the ED, and not practices by individuals who were treated in the ED.

Thank you for this comment, we have amended the sentence as you suggest, which can now be found on Page 3, Line 85, and is pasted here:

The objective of this study was to assess mask wearing beliefs and practices, specifically mask use, access to masks, and reasons for not wearing masks, among patients treated in emergency departments (EDs) that serve as critical points of health care access and intervention for several vulnerable populations. 

Since not all sites conducted the survey in the same manner (some while in the ED and some after discharge), this may be an additional limitation to note. Subjects may respond differently in different situations.

Thank you for this comment, we have amended the sentence as you suggest, which can now be found on Page 11, Line 221, and is pasted here:

Three sites used different methods for the survey (two via telephone during ED stay and one via telephone after ED discharge), formats which may lead to different answers to key questions.

Not sure about the grouping of responses for mask wearing outside the home. This may be ok but would be helpful to include an additional table that shows frequency and percentage for each response of always, most of the time, sometimes and never. Always (100%) is quite different from most of the time (50-99%?). 

Thank you for this excellent suggestion. We have provided this information through the addition of an additional table, included in the supplement. This table describes the demographic information by each of the four mask wearing categories. This table is included in the supplement.

Were respondents provided the percentages (> or < 50% of the time) to help quantify their responses?

Thank you for this comment. Respondents were provided with percentages to help quantify their responses. We have amended the manuscript to make this clearer, which can now be found on Page 5, Line 126, and is pasted here:

“Do you wear a mask when you are outside of your home when you are around other people?” with answer choices a) always, b) most of the time (more than 50%), c) sometimes, but less than half of the time (less than 50%), and d) I never wear a mask. Respondents were provided with these percentages to help quantify their responses. 

The reported mask wearing in the results is confusing. Again, would be helpful to provide numbers for each response and then group after that.

Thank you again for this suggestion. We have included this information in the supplemental table pasted above.

Survey item number 12 regarding regular clinic or doctor does not specifically state primary care doctor. Is it assumed that respondents understand this item meant primary care doctor? How would it differentiate if they saw a specialist regularly?

Thank you for this comment. It was our feeling that the term primary care doctor would be somewhat ‘jargon’ term for our populations and that they would understand regular doctor as their primary care provider. We have used this exact wording/question in other published studies of similar design. (Rodriguez RM et al. The Rapid Evaluation of COVID-19 Vaccination in Emergency Departments for Underserved Patients Study. Ann Emerg Med 2021;78:502–10. https://doi.org/10.1016/J.ANNEMERGMED.2021.05.026.) 

The health insurance survey response options are confusing. Did all 15 participating sites outside of the lead site offer MediCal, Kaiser or Healthy San Francisco? Each site probably has a similar version of these programs. Is Kaiser considered private insurance? How this data is reported should be clarified and may require to be grouped differently. Could also consider an overall grouping of insured and not insured and then provide the breakdown of different types.

Thank you for this excellent comment. The survey submitted in our supplement is the version used at the lead site. Each of the remaining sites revised their survey to include wording applicable to their community (i.e. the site in Los Angeles changed Healthy SF to Healthy Los Angeles), and these local community health plans were coded together. We have updated Table 1 to reflect this by changing the Healthy SF designation to Local Community Health Plan. Kaiser health plan was available in many communities outside of California – including in Seattle and remained on the survey. No other options were added by other sites. We have amended the manuscript as you suggest, which can now be found on Page 5, Line 117. We have also amended the Table to include a line first of insured vs uninsured and then broken out by groups. 

The survey submitted in our supplement (S1) is the version used at the lead site. Each of the remaining sites revised their survey to include wording applicable to their community (i.e., the site in Los Angeles changed Healthy San Francisco to Healthy Los Angeles), and these local community health plans were coded together.

On Table 1, results are presented in different ways but it is unclear which is what. Some are median and IQR, some appear to be frequency and percentage and others a range? Ok to report differently but each needs to be stated what is the measure. Otherwise it is confusing.

Thank you for this excellent comment. We have amended the Table to include the differences in reporting for each result.

For the supplemental survey items, would consider only including the items that were relevant to the reported results for this study. For example, this study did not report on influenza or COVID vaccinations yet there are survey items included on this.

Thank you for this excellent comment. We have amended the supplement as suggested and removed questions not pertaining to this study.

The Discussion could be more robust. More specifics on how this study compares to similar studies should be included. Should elaborate more on the issues presented, especially as related to vulnerable populations. Discussion should note the implications for clinical practice and areas for future or additional research.

Thank you for this comment. We have amended the discussion as suggested, which can now be found on Page 10, Line 194 and pasted below.

In this real-world ED study of diverse, urban populations, respondents reported a high rate of facemask use. These rates are higher than some which have been previously reported.[16] However, the sources of masks and reasons for not wearing masks were similar to those found in other research.[11] Small but significant differences in mask wearing habits were found between those of different racial/ethnic groups, and between males and females, differences which have been found in prior surveys as well.[17] Our study adds to these data due to the unique population studied, patients in safety-net EDs. As such we were able to note that individuals experiencing homelessness were significantly less likely to wear masks most of the time than those who are housed. This potentially points to an unmet need amongst this population to provide education and supplies. With businesses such as coffeehouses closed, it is possible that access to electrical outlets was limited, making it difficult to charge mobile phones and other devices, and thus essential information regarding social distancing and protective measures may not have been easily transmitted to this group.[18] Mask wearing may have also been limited by inabilities to keep masks clean and dry or limited access to masks at food banks and shelters, especially early in the pandemic. [19]

 The primary limitation of this study is that we conducted this survey in December 2020 through February 2021 – notably not during the Delta or Omicron variant surges. With progressive “pandemic fatigue”[20] and decreased mask mandates,[21] mask wearing practice may have changed. Because this data is self-reported, social desirability bias may have influenced respondents to answer in ways they thought may please survey administrators, even if not representative of their true behavior. Spectrum bias limits the generalizability of our findings: all sites were in urban areas affiliated with academic medical institutions, and we could only survey people who presented to EDs, thus excluding patients in rural areas, where mask use is known to be less prevalent,[22,23] and those who may have greater mistrust of health care systems. We did not separately assess race and ethnicity (Supplement) and thus it is possible that we did not accurately capture these demographics, especially for Latinx individuals. Nuances of mask use, such as correct positioning, were not assessed, and thus our results may overestimate effective mask wearing. Three sites used different methods for the survey (two via telephone during ED stay and one via telephone after ED discharge), formats which may lead to different answers to key questions. 

Despite these limitations, this study delineates an opportunity for public health intervention from the ED – one that should not take significant time or cost. Our study shows that currently very few respondents received masks from the ED or their doctors’ offices. Emergency departments should consider social determinants of health as they relate to the provision of masks to vulnerable populations, especially people experiencing homelessness,[24,25] ED providers should ask patients whether they wear masks and if they have enough of them, as well provide education regarding their use. Educating patients about the benefits of masks and providing masks to those who do not have enough should become part of provider care in the ED – perhaps as a standard component of triage or discharge. Further research, with more contemporary data, to assess the feasibility and acceptability among patients and providers of such interventions should be done to guide next steps.

The timeframe in which the study occured is stated but should be specifically noted it did not include the Delta or Omicron variant surges.

Thank you for this comment. We have amended the manuscript as you suggest, which can now be found on Page 11, Line 210, and is pasted here:

The primary limitation of this study is that we conducted this survey in December 2020 through February 2021 – notably not during the Delta or Omicron variant surges.

Additional study limitation is subject self reported data. There is always the possibility that respondents may not have answered truthfully but rather based on what they felt the healthcare team wanted to hear.

Thank you for this comment. We mentioned social desirability bias as a limitation, have amended the manuscript to make this clearer, which can now be found on Page 11, Line 212, and is pasted here:

Because this data is self-reported, social desirability bias may have influenced respondents to answer in ways they thought may please survey administrators, even if not representative of their true behavior

The Conclusion section should be more concise. This is typically a short summary of the overall findings. The second part of the conclusions regarding actionalble items and education belong in the discussion. Conclusion should not note anything new that wasn't already discussed.

Thank you for this comment. We have shortened the conclusion section by moving our sentence on actionable items to the discussion section. The amended conclusion section can be found on Page 12, Line 235, and is pasted here:

Our findings inform interventions in the ED focusing on homeless persons and those without PCPs, who had significantly lower rates of mask wearing and greater difficulty obtaining masks. Having a PCP may provide opportunities for education and distribution of facemasks, though it is also possible that accessing primary care is correlated with greater health care access and economic security, and thus greater access to masks.

Regarding the figure of percentage of respondents by hospital, it would also be helpful to include something similar with grouping by geographical region.

Thank you for this comment. We have amended Figure 1 to include Hospital names and state to provide information on geographical region and have pasted the updated figure here:

Overall the study is good and I commend the authors for their work. Improving on the points noted will help it to be more meaningful and contribute to the literature on this topic.

Reviewer #2: Thank you very much for allowing me to review the submission titled: Facemasks: Perceptions and Use in an ED Population During COVID-19, submitted by Karen Adams et all. This paper is a secondary analysis of an ED based study across the US focusing on safety net EDs in urban academic institutions. Their data is interesting but biased by where the study was run.

While they acknowledge that bias in their conclusion, throughout the result section we are able to see the degree of imbalance in their population studied. Their data shows that English was the primary language for 80%, with 81% having a PCP and only 12% being uninsured. By documenting that the white, non Latinx population buys their masks on line, we are left with the suggestion that these are adults of means, with access to on line ordering, a home/stable place of living for product to be delivered to and a credit card to pay the bill. In essence that does not translate to their results being extendable to match what the title of their study is. In general, their population is able to acquire masks and uses them as needed. Only a very few do not want or use masks, and even fewer document that they don’t want to use them. This does not match the large general population that are anti-mask, that uses social media to downplay the risks of COVID and play up the myth that masks are bad for you.

Perhaps we would have been better able to judge mask use if we looked at the general population that attended large (political) gatherings or casinos or even looked in states that never had a mask mandate.

Thank you for this comment, however, we strongly disagree with these assertions. Our population of study was the exact population that receives care in urban EDs. With 15 sites across the country, we have a broad sampling including sites in the northeast, south, Midwest, California, and the Northwest. The goal of this study was not to look specifically anti mask populations but rather to look at populations served by safety net EDs, as is stated in our objection section. Additionally, with regards to the source of masks, only 13% overall obtained masks online and only 18% of White non-Latinx, 10% Black non Latinx and 8% of Latinx patients obtained masks online – a minority and thus not generalizable to the entire population.

---

## [Editor Report · Decision Letter 1]

15 Mar 2022

Facemasks: Perceptions and Use in an ED Population During COVID-19

PONE-D-22-02590R1

Dear Dr. Eswaran,

We’re pleased to inform you that your manuscript has been judged scientifically suitable for publication and will be formally accepted for publication once it meets all outstanding technical requirements.

Kind regards,

Francesca Baratta, PharmD, PhD

Academic Editor

PLOS ONE

---

## [Editor Report · Acceptance letter]

24 Mar 2022

PONE-D-22-02590R1 

Facemasks: Perceptions and Use in an ED Population During COVID-19 

Dear Dr. Eswaran:

I'm pleased to inform you that your manuscript has been deemed suitable for publication in PLOS ONE. Congratulations! Your manuscript is now with our production department. 

Kind regards, 

on behalf of

Dr. Francesca Baratta 

Academic Editor

PLOS ONE